# RECONSTRUCTIONNET: A NEURAL NETWORK ARCHITECTURE FOR UNCERTAINTY-AWARE PREDICTIONS WITH EXPLAINABILITY

## ABSTRACT

Uncertainty estimation quantifies a model's confidence in its predictions, fostering calibrated trust among users. Existing approaches face two key limitations: (1) most capture only a single type of uncertainty, and (2) they incur additional training or inference overhead. We propose ReconstructionNet, a neural network that addresses these limitations by modeling the joint input–output distribution with class-specific autoencoders. This enables simultaneous prediction and estimation of both aleatoric and distributional uncertainty in a single pass. Across five real-world datasets, ReconstructionNet matches or surpasses baseline classifiers while producing uncertainty estimates with greater reliability, selectivity, robustness to false negatives, and strong out-of-distribution detection. Furthermore, ReconstructionNet's architecture naturally supports uncertainty explanations, revealing how individual features contribute to prediction uncertainty without extra computation. Experiments demonstrate that these explanations highlight misclassified regions consistent with human intuition. Together, these contributions establish ReconstructionNet as a unified framework for trustworthy and interpretable artificial intelligence.

## 1 INTRODUCTION

Uncertainty estimation refers to the task of quantifying how uncertain a machine learning model is about its prediction for each instance. Reliable uncertainty estimates foster calibrated trust by alerting users to cases where the model is likely to be uncertain and erroneous (Toh et al., 2025). Common methods for uncertainty estimation include Bayesian Neural Networks (BNNs) (Jospin et al., 2022), Monte Carlo Dropout (Gal & Ghahramani, 2016) and Deep Ensemble (Lakshminarayanan et al., 2017). These methods quantify one or more of three main types of uncertainty (Malinin & Gales, 2018): 1) Aleatoric (data), 2) Epistemic (model), and 3) Distributional uncertainty. While existing work in uncertainty estimation shows promise, it often faces several limitations. Most uncertainty estimates quantify only a single type of uncertainty and are unable to differentiate between various sources of uncertainty. Furthermore, many uncertainty estimation methods incur increased training and inference time.

We introduce **ReconstructionNet**, a neural network architecture designed to address the aforementioned limitations. ReconstructionNet quantifies aleatoric and distributional uncertainty by modeling the joint input–output distribution with class-specific autoencoders, effectively measuring the distance of an instance from the training data of each class. This design reduces epistemic uncertainty by constraining the state space and allows ReconstructionNet to distinguish between aleatoric and distributional uncertainty.

Beyond identifying when a model is uncertain, understanding which features contribute to prediction uncertainty is equally valuable, giving rise to the emerging field of uncertainty explanation (Wang et al., 2025; Antorán et al., 2021; Fan, 2025). Uncertainty explanations are typically represented as vectors of real values, where each value quantifies the significance and impact of an input feature on the model's uncertainty. State-of-the-art methods include applying out-of-the-box eXplainable Artificial Intelligence (XAI) techniques, such as Integrated Gradients (IG) (Sundararajan et al., 2017), to explain existing uncertainty estimates (Mougan & Nielsen, 2023; Iversen et al., 2024). Other

approaches involve observing how input perturbations affect prediction uncertainty (Antorán et al., 2021; Wang et al., 2025) and learning which features significantly reduce uncertainty. However, these methods require extra modules on top of uncertainty estimation, increasing inference time.

ReconstructionNet's design enables inbuilt uncertainty explanation: each class-specific autoencoder produces feature-wise reconstruction errors scaled by learned error weights for classification. Those weighed reconstruction errors quantify each feature's contribution to the uncertainty, providing uncertainty explanations without additional modules or computation.

The contributions of this research are as follows:

1. Propose ReconstructionNet, a neural architecture that minimises epistemic uncertainty while quantifying and explaining both aleatoric and distributional uncertainty.

2. Provide a theoretical evaluation of ReconstructionNet's uncertainty explanations.

3. Demonstrate the efficacy of ReconstructionNet for prediction, uncertainty estimation, and explanation on real-world applications in healthcare and finance.

## 2 RELATED WORK

### 2.1 UNCERTAINTY ESTIMATION

**Definition 1 (Uncertainty Estimation)** *For instance $\mathbf{x} \in \mathbb{R}^d$ and a model $f$, an uncertainty estimator $\sigma(\mathbf{x}; f) : \mathbb{R}^d \to \mathbb{R}$ assigns a real-valued measure of the prediction uncertainty for $\mathbf{x}$.*

Uncertainty estimates generally aim to quantify three types of uncertainty:

1. **Aleatoric**: Uncertainty arising from inherent noise in the training data. In classification problems, this manifests as overlapping classes.

2. **Epistemic**: Uncertainty from inadequate model parameter fit, reducible by expanding the dataset or narrowing the hypothesis space.

3. **Distributional**: Uncertainty caused by data shifts between the training and prediction set.

**Bayesian methods** capture epistemic uncertainty by modeling distributions over network weights. For instance, Bayesian Neural Networks (BNNs) (Jospin et al., 2022) sample from these distributions to produce multiple predictions, with the variability reflecting uncertainty. Deep Ensembles (Lakshminarayanan et al., 2017) and Monte Carlo Dropout (MCD) (Gal & Ghahramani, 2016) approximate this sampling via multiple models or stochastic weight activations within a single network. More recently, efficient Bayesian last-layer approaches such as Variational Bayesian Last Layers (Harrison et al., 2024) and Bayesian Non-negative Decision Layers (Hu et al., 2025) reduce computational cost by restricting stochasticity to the final layer. While Bayesian methods remain powerful, they typically require additional forward passes compared to deterministic models.

**Evidential Deep Learning (EDL)** methods (Ulmer et al., 2023; Amini et al., 2020) are more computationally efficient than Bayesian approaches, requiring only a single forward pass and one trained model. They assume the output follows a well-characterized distribution, predicting the parameters of the output distribution from which uncertainty can be derived. For classification, this corresponds to Dirichlet parameters (Sensoy et al., 2018), with Posterior Networks (PN) (Charpentier et al., 2020) extending this via normalizing flows. Despite capturing aleatoric and distributional uncertainty, these methods rely on restrictive assumptions about the output distribution.

**Deterministic Uncertainty Methods (DUMs)** (Postels et al., 2022; Charpentier et al., 2023; Zelenka et al., 2023a) make minimal assumptions about the output distribution, estimating distributional uncertainty as the distance of an instance from the training set. A recent DUM, the Reconstruction Uncertainty Estimate (RUE) (Wang et al., 2024; Korte et al., 2024), uses a decoder to reconstruct inputs from the latent representation, with reconstruction error serving as the uncertainty measure. ReconstructionNet extends RUE by also capturing aleatoric uncertainty alongside distributional uncertainty.

## 2.2 UNCERTAINTY EXPLANATIONS

Knowing when a model is unreliable is valuable, but uncertainty explanations provide deeper insight by quantifying the contribution of each feature to the model's overall uncertainty.

**Definition 2 (Uncertainty Explanation)** *Given an instance $\mathbf{x} \in \mathbb{R}^d$, a model $f$, and an uncertainty estimator $\sigma(\mathbf{x}; f) : \mathbb{R}^d \to \mathbb{R}$, an uncertainty explanation method $\zeta(\mathbf{x}; f, \sigma) : \mathbb{R}^d \to \mathbb{R}^d$ assigns each input feature a real value reflecting its contribution to $f(\mathbf{x})$'s uncertainty.*

Uncertainty explanation is a nascent field with two primary approaches:

**Gradient-Based Methods** (Mougan & Nielsen, 2023; Iversen et al., 2024) applied standard eXplainable Artificial Intelligence (XAI) methods, such as Integrated Gradients (IG) (Sundararajan et al., 2017), to explain uncertainty estimates. While easy to implement, gradient-based explanations can be sparse due to vanishing gradients, making them difficult to interpret.

**Perturbation-Based Methods** (Antorán et al., 2021; Wang et al., 2025) assess each feature's contribution to uncertainty by perturbing inputs and measuring the impact on the uncertainty score. Their accuracy depends on the number of perturbations, making them computationally expensive.

Both methods require an additional module for uncertainty explanation, increasing inference time. In contrast, (Zelenka et al., 2023b) computes predictions, uncertainty, and explanations in a single forward pass using one model. Based on prototype networks (Snell et al., 2017), it classifies instances by similarity to class prototypes, with uncertainty explanations derived from the inner product between the predicted prototype and the instance's feature map. Similarly, ReconstructionNet leverages its architecture to provide ante-hoc uncertainty explanations efficiently during inference.

## 2.3 RECONSTRUCTION-BASED METHODS

**Anomaly Detection.** Reconstruction error has been widely used in autoencoder-based anomaly detection Chen et al. (2018) to identify deviations from the training distribution. While ReconstructionNet also leverages reconstruction errors, it differs substantially in several aspects:

*Reconstruction for Classification and Uncertainty Estimation.* Traditional anomaly-detection autoencoders detect out-of-distribution (OOD) inputs only. In contrast, ReconstructionNet uses class-specific autoencoders to model the joint input-output distribution and classifies by selecting the class with the lowest reconstruction error. This reconstruction error also serves as a distributional uncertainty estimate. Reconstruction thus supports both classification and uncertainty estimation in ReconstructionNet.

*Reconstruction Errors as Explanations.* Reconstruction errors in ReconstructionNet serve as feature-level uncertainty explanations, highlighting which input features drive uncertainty, unlike conventional anomaly detectors that produce only scalar anomaly scores.

*Training Objective and Architecture.* Unlike unsupervised anomaly detection, ReconstructionNet uses a supervised training objective and class-specific autoencoders, where each class has its own encoder-decoder pair. This design enables joint modeling of input-output distributions.

**Reconstruction Error as Regularizers** Some works use reconstruction loss as a regularizer for classification Le et al. (2018); Ghifary et al. (2016). ReconstructionNet differs fundamentally in several aspects:

*Class-specific autoencoders and joint distribution modeling.* Prior works use a single autoencoder with conditional probabilities for classification. ReconstructionNet employs one autoencoder per class to model the joint input-output distribution, classifying via minimum reconstruction error.

*Uncertainty estimation and explanations.* Unlike prior works, ReconstructionNet leverages reconstruction errors to quantify uncertainty and provide feature-level explanations.

*Distinct architecture and prediction mechanism.* Instead of connecting encoders directly to the prediction head, ReconstructionNet computes weighted reconstruction errors across class-specific autoencoders to make predictions.

## 3 METHODOLOGY

We present ReconstructionNet, a neural network architecture which offers the following features:

1. Uses the joint input-output probability for classification.
2. Quantifies distributional and aleatoric uncertainty.
3. Generates uncertainty explanations.

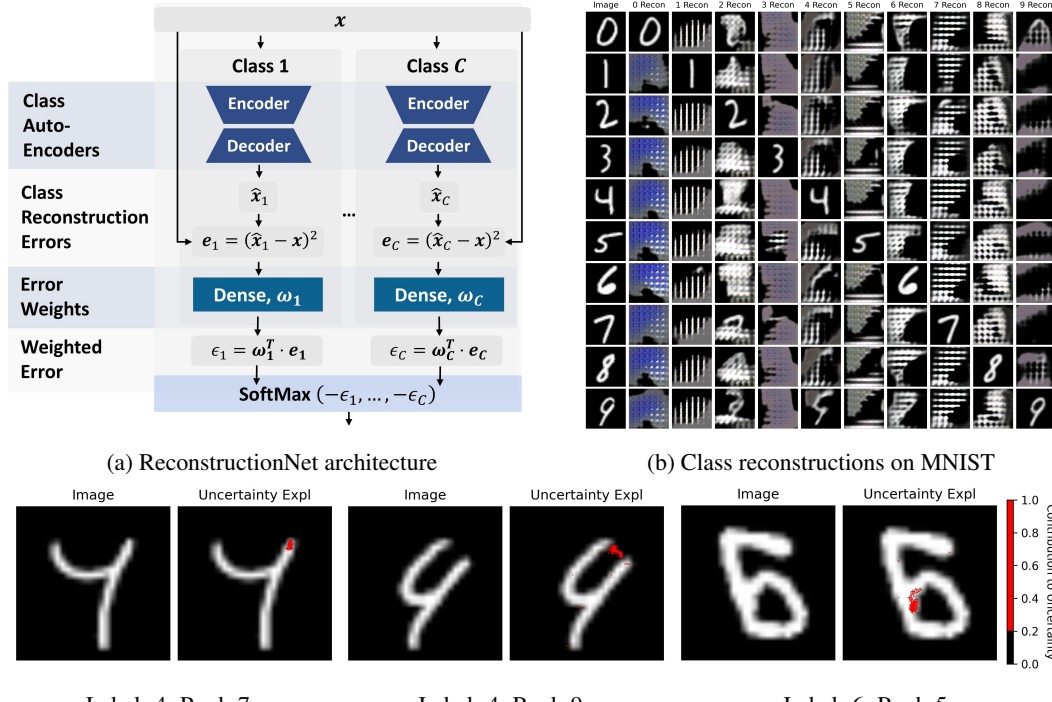

(a) ReconstructionNet architecture

(b) Class reconstructions on MNIST

(c) ReconstructionNet uncertainty explanations on MNIST

Figure 1: ReconstructionNet overview and examples. (a) ReconstructionNet architecture: Each class-specific autoencoder is trained to reconstruct only its own class, producing reconstruction errors inversely related to the joint input–output probability. After applying error weights and softmax normalization, the resulting probabilities are used for classification (Equation 2). (b) MNIST class reconstructions: Only the autoencoder for the true class yields a faithful reconstruction, leading to lower reconstruction error and a higher prediction probability. Other autoencoders generate artifacts (e.g., checkerboard patterns (Odena et al., 2016)) due to lack of training on mismatched classes. (c) MNIST uncertainty explanations: Weighted class reconstruction errors highlight uncertain regions. In the first example, the extra right vertical line increases uncertainty of it being a 7; in the middle, the missing connector raises uncertainty of it being a 9; in the last, the extra bottom-left vertical line increases uncertainty of it being a 5. See Appendix A.4 for implementation details.

### 3.1 CLASSIFICATION

Consider a classification dataset with $N$ instances and $C$ classes, where each instance $i$ has an input vector $\mathbf{x}_i \in \mathbb{R}^d$ and label $y_i$. Let $\mathcal{X}_j$ denote the set of training instances with target label $j$, containing $N_j$ instances. For each instance $i$ and class $j$, the true and predicted class probabilities are $p_{ij}$ and $\hat{p}_{ij}$, respectively.

Traditional neural networks predict the conditional probability of class $j$ given instance $\mathbf{x}_i$. The class with the highest conditional probability is then selected as the final predicted class $\hat{y}_i$:

$$\hat{y}_i = \underset{j \in \{1,\dots,C\}}{\arg\max} \Pr\left(\hat{y}_i = j \mid \mathbf{x}_i\right). \tag{1}$$

ReconstructionNet differs from traditional feedforward neural network classifiers in its inference process. ReconstructionNet instead predicts the joint probability of the target $\hat{y}_i$ and input $\mathbf{x}_i$ and the target class with the highest joint probability is the final predicted class (Equation 2):

$$\hat{y}_i = \underset{j \in \{1, \ldots, C\}}{\arg \max} \Pr\left(\hat{y}_i = j, \mathbf{x}_i\right). \tag{2}$$

Equation 2 is a valid classification formulation, equivalent to Equation 1 via Bayes' theorem:

$$\underset{j}{\arg \max} \Pr\left(\hat{y}_i = j, \mathbf{x}_i\right) = \underset{j}{\arg \max} \Pr\left(\hat{y}_i = j \mid \mathbf{x}_i\right) \Pr(\mathbf{x}_i) = \underset{j}{\arg \max} \Pr\left(\hat{y}_i = j \mid \mathbf{x}_i\right). \tag{3}$$

To model joint input-output probability, we adopt the model architecture in Figure 1a. Given a classification problem with $C$ classes, we construct $C$ autoencoders $g_1, \ldots, g_C$ and train them simultaneously, such that each autoencoder models the joint probability $\Pr(\hat{y}_i = j, \mathbf{x}_i)$. During inference, the model computes feature-wise reconstruction errors $\mathbf{e}_{ij}$ for each autoencoder (Equation 4). $\hat{\mathbf{x}}_{ij} = g_j(\mathbf{x}_i)$ is the reconstructed input of instance $i$ by class-$j$ autoencoder $g_j$; $\hat{\mathbf{x}}_{ij}^k$ is the reconstruction for feature $k$.

$$\mathbf{e}_{ij} = \left[\left(\hat{\mathbf{x}}_{ij}^1 - \mathbf{x}_i^1\right)^2 \quad \cdots \quad \left(\hat{\mathbf{x}}_{ij}^d - \mathbf{x}_i^d\right)^2\right] \tag{4}$$

Next, it calculates a weighted reconstruction error $\epsilon_{ij}$ for each class $j$, where the weights $\omega_j$ are trainable parameters (Equation 5).

$$\epsilon_{ij} = \omega_j^{\mathsf{T}} \cdot \mathbf{e}_{ij}. \tag{5}$$

Finally, prediction probability $\hat{p}_{ij}$ is obtained by applying softmax to the negative weighted errors $\epsilon_i$. Negation is used since the higher the reconstruction error, the lower the probability (Figure 1b).

$$\hat{p}_i = \text{softmax}\left(-\boldsymbol{\epsilon}_i\right). \tag{6}$$

To train the ensemble of class autoencoders concurrently, we designed a loss function (Equation 7) consisting of two components, with $\beta$ as a hyperparameter to balance both training objectives.

$$\mathcal{L}_{total} = \mathcal{L}_{CE} + \beta \cdot \mathcal{L}_{Class\_MSE}, \quad \mathcal{L}_{CE} = -\sum_{i=1}^{N} \sum_{j=1}^{C} p_{ij} \log(\hat{p}_{ij}) \tag{7}$$

$\mathcal{L}_{CE}$ is the cross-entropy loss; it optimizes the classification performance of the ReconstructionNet. To ensure that the class autoencoders also learn to model the joint probability $\Pr(\hat{y}_i = j, \mathbf{x}_i)$ while also maintaining predictive accuracy, we introduce the class-dependent mean squared error (MSE):

$$\mathcal{L}_{Class\_MSE}^{j} = \frac{1}{N_j} \sum_{\mathbf{x}_i \in \mathcal{X}_j} \|\hat{\mathbf{x}}_{ij} - \mathbf{x}_i\|^2. \tag{8}$$

Where each class autoencoder $g_j$ is trained to minimize reconstruction error for its corresponding instances in $\mathcal{X}_j$. The overall reconstruction loss, $\mathcal{L}_{Class\_MSE}$, is then the average across all classes:

$$\mathcal{L}_{Class\_MSE} = \frac{1}{N} \sum_{j=1}^{C} N_j \cdot \mathcal{L}_{Class\_MSE}^{j}. \tag{9}$$

The formulation of $\mathcal{L}_{Class\_MSE}$ ensures each class autoencoder is trained exclusively to reconstruct samples from its ground truth class, thereby modeling the joint distribution of the input and target. Assume that the latent vector $z_j$ from the encoder of class $j$ follows an isotropic Gaussian distribution, $\Pr(\mathbf{x}_i, y_i = j \mid z_{ij}) = \mathcal{N}(\hat{\mathbf{x}}, \sigma^2 I)$ (Doersch, 2021; Odaibo, 2019). The probability density function of the distribution of all $\mathbf{x}$ from class $j$ can be expressed as:

$$\Pr(\mathbf{x}_i, y_i = j \mid z_{ij}) = \frac{1}{(2\pi\sigma^2)^{d/2}} \exp\left(-\frac{1}{2\sigma^2} \|\mathbf{x}_i - \hat{\mathbf{x}}_{ij}\|^2\right). \tag{10}$$

To model this distribution, we optimize our model to maximize the log-likelihood over the training dataset with target class $j$:

$$\ell(\hat{\mathbf{x}}, \sigma) = \sum_{i \in \mathcal{X}_j} \left[-\frac{d}{2} \log(2\pi\sigma^2) - \frac{1}{2\sigma^2} \|\mathbf{x}_i - \hat{\mathbf{x}}_{ij}\|^2\right] \tag{11}$$

$$= \sum_{i \in \mathcal{X}_j} \left[-\frac{d}{2} \log(2\pi\sigma^2)\right] - \frac{N_j}{2\sigma^2} \cdot \mathcal{L}_{Class\_MSE}^{j}. \tag{12}$$

We observe that maximizing the log-likelihood is equivalent to minimizing $\mathcal{L}_{Class\_MSE}^{j}$. This demonstrates that minimizing $\mathcal{L}_{Class\_MSE}$ leads to the training of class autoencoders that model the joint distribution of the input and target, with reconstruction error inversely related to the joint probability. Additionally, the architecture and loss function of ReconstructionNet limit its hypothesis space, making it more resistant to epistemic uncertainty (Hüllermeier & Waegeman, 2021).

## 3.2 Uncertainty Estimation

By modelling the joint input-output probability, we can quantify:

1. **Aleatoric Uncertainty**: When instances lie in overlapping class regions, several classes have similarly *high* probabilities above threshold $\theta_1$, signalling high aleatoric uncertainty.

$$\exists \, \mathbf{C}_i \subseteq \{1, \dots, C\}, \ |\mathbf{C}_i| \geq 2 \ \text{ s.t.}$$
$$\Pr(\hat{y}_i = c_1, \mathbf{x}_i) \approx \Pr(\hat{y}_i = c_2, \mathbf{x}_i) \geq \theta_1, \ \forall \, c_1, c_2 \in \mathbf{C}_i. \tag{13}$$

2. **Distributional Uncertainty**: When instances lie beyond the training distribution, the joint probabilities of all classes are similarly *low*, below some threshold $\theta_2$, signaling high distributional uncertainty.

$$\forall \, c \in \{1, \dots, C\}, \ \Pr(\hat{y}_i = c, \mathbf{x}_i) \leq \theta_2. \tag{14}$$

Modelling joint probabilities allows differentiation between uncertainty types: instances show high aleatoric uncertainty when their most probable classes have probabilities above $\theta_1$, and high distributional uncertainty when all class probabilities are below $\theta_2$.

The notion of aleatoric uncertainty, as illustrated in Equation 13, is nicely captured by Shannon entropy, reflecting evenly spread high prediction probabilities across overlapping classes:

**Definition 3 (Aleatoric Uncertainty)** *For an instance* $\mathbf{x}_i$, *aleatoric uncertainty is quantified using the Shannon entropy (Shannon, 1948) of the prediction probabilities* $\hat{p}_{ij}$:

$$\sigma_{aleatoric}(\mathbf{x}_i) = -\sum_{j=1}^{C} \hat{p}_{ij} \log \hat{p}_{ij}.$$

Distributional uncertainty in Equation 14 is captured by the predicted class's reconstruction error. By design, the predicted class has the lowest error and highest probability among all classes; therefore, if its error is high (probability low), all other classes also have low probabilities.

**Definition 4 (Distributional Uncertainty)** *For an instance* $\mathbf{x}_i$, *distributional uncertainty is the reconstruction error of the predicted class* $\hat{y}_i$'s *autoencoder:*

$$\sigma_{dist}(\mathbf{x}_i) = \|\mathbf{e}_{i\hat{y}_i}\|_1.$$

## 3.3 Uncertainty Explanation

The weighted reconstruction errors $\zeta$ of the predicted class $\hat{y}$ serve as uncertainty explanations, as they represent feature uncertainties scaled by their importance to the prediction.

**Definition 5 (ReconstructionNet Explanation)** *The* ReconstructionNet Explanation *for instance* $\mathbf{x}_i$ *and its predicted class* $\hat{y}_i$ *is the weighted reconstruction errors of predicted class* $\hat{y}_i$:

$$\zeta(\mathbf{x}_i) = \omega_{\hat{y}_i} \, \odot \, \mathbf{e}_{i\hat{y}_i}.$$

ReconstructionNet uncertainty explanations exhibit the following three properties: (1) Implementation Invariance (Sundararajan et al., 2017), (2) Sensitivity (Sundararajan et al., 2017), and (3) Consistency (Lundberg & Lee, 2017). An explanation that satisfies all three properties is (1) consistent across different implementations, (2) does not attribute irrelevant features incorrectly, and (3) preserves the relative importance of features across models.

An uncertainty explanation is *implementation-invariant* if, for a pair of functionally equivalent prediction models, the same explanation is generated for any instance. A pair of functionally equivalent models are models that yield the same output for a given set of inputs.

**Property 1 (Implementation Invariance)** *Given two functionally equivalent prediction models, $f$ and $f'$, an explanation function $\chi$ is implementation-invariant if and only if, for any instance $\mathbf{x}$, the explanations derived from both models using $\chi$ are equivalent: $\chi(\mathbf{x}; f) = \chi(\mathbf{x}; f')$.*

ReconstructionNet's uncertainty explanations $\zeta$, defined as the weighted reconstruction errors of the predicted class (Definition 5), are implementation-invariant: for two functionally equivalent models, both the error weights $\omega$ and the reconstruction errors remain identical.

A *sensitive* explanation function allocates zero attribution to irrelevant features for prediction. A feature is considered irrelevant if a change in its value does not impact the model's prediction.

**Property 2 (Sensitivity)** *An explanation function $\chi$ is sensitive if it assigns a zero feature attribution value, $\chi(\mathbf{x}; f)_i = 0$, to features $i$ that are irrelevant to the prediction.*

ReconstructionNet's uncertainty explanations $\zeta$ are sensitive. Since the error weight $\omega$ encodes each feature's contribution to the final prediction, any change in feature uncertainty that does not affect the prediction must have a weight of zero, yielding a weighted reconstruction error $\zeta$ of zero.

An explanation is *consistent* if a feature's uncertainty attribution does not decrease when the model is altered to increase that feature's contribution.

**Property 3 (Consistency)** *Let $f'$ be a modification of $f$ where feature $i$'s contribution is increased. For an instance $\mathbf{x}$ and $\mathbf{x}^{\backslash i}$ with $x_i = 0$, an explanation $\chi$ is consistent if:*

$$f'(\mathbf{x}) - f'\left(\mathbf{x}^{\backslash i}\right) \geq f(\mathbf{x}) - f\left(\mathbf{x}^{\backslash i}\right) \quad \text{then} \quad \chi\left(\mathbf{x}; f'\right)_i \geq \chi\left(\mathbf{x}; f\right)_i.$$

ReconstructionNet's uncertainty explanations, $\zeta$, are consistent. For any instance, reconstruction errors stay the same between $f$ and $f'$, since the joint input-output probability modeled by the autoencoders is independent of feature contributions. Thus, only the error weights $\omega$ can change; if a feature's weight increases in $f'$, its uncertainty attribution also increases, satisfying consistency.

## 4 EXPERIMENTS

### 4.1 DATASETS

We use tabular datasets (Covid, Diabetes, Fund) to evaluate uncertainty reliability, selectivity, and robustness, medical image datasets (ISIC, OCTMNIST) for OOD detection, and MNIST, ISIC, Covid and Diabetes to assess uncertainty explanation correctness. The datasets are summarized as follows:

1. **Covid** (Hinns et al., 2021) is a tabular dataset of United Kingdom's COVID-19 policies and regional case counts, labelled by whether $R_t$ (effective reproduction number) $> 1$.

2. **Diabetes** (Mustafa, 2023) is a binary tabular dataset of demographics, pre-existing conditions, and vital signs, with labels indicating diabetes status.

3. **Fund** (Kovvuri et al., 2023) is a binary tabular dataset from 4,330 funds, using macroeconomic indicators, fund allocations, HHI, and past performance to predict if a fund's net asset value (NAV) exceeds the previous quarter's.

4. **ISIC** (Codella et al., 2019; Tschandl et al., 2018) is a dermoscopic image dataset of seven skin conditions. For OOD detection, we created three datasets with decreasing similarity to ISIC: BCN-IN (images from seen classes of BCN20000 (Combalia et al., 2019)), BCN-OUT (images from the unseen Scar class), and ChestMNIST (Wang et al., 2017).

5. **OCTMNIST** (Kermany et al., 2018) is a retinal OCT dataset with four classes. For OOD detection, we used three datasets of decreasing similarity to OCTMNIST: OCTDL-IN (images from seen classes of OCTDL (Kulyabin et al., 2024)), OCTDL-OUT (images from unseen classes), and ChestMNIST (Wang et al., 2017).

6. **MNIST** (Deng, 2012) is an image dataset of ten handwritten digits.

## 4.2 BASELINES

We compared ReconstructionNet against six recent state-of-the-art uncertainty estimation methods: (1) Entropy, (2) MCD, (3) DE, (4) PN, (5) BNN, and (6) EDL. Implementation and tuning details of the baselines are provided in the appendix.

## 4.3 EVALUATION METRICS

We evaluate prediction performance with Area Under receiver operating characteristic Curve (AUC) and accuracy. To evaluate uncertainty estimation performance, we used the following metrics:

1. **Correlation** (Mi et al., 2022; Upadhyay et al., 2022) measures reliability as the Pearson correlation between uncertainty and error, with higher values indicating better reliability.

2. **AURC** (Ding et al., 2020) quantifies selectivity by measuring the area under the risk-coverage curve (AURC), with lower values indicating better selectivity.

3. $\sigma$-**Risk Score** measures robustness to false negatives as errors for instances with normalized uncertainty below $\sigma = \{0.1, 0.2, 0.3, 0.4\}$, with lower values indicating greater resilience.

4. **OOD Detection** (Lakshminarayanan et al., 2017; Postels et al., 2020; Malinin & Gales, 2018) measures how well uncertainty distinguishes in-distribution from OOD instances using AUROC, with higher values indicating better detection.

## 4.4 RESULTS

### 4.4.1 PREDICTION PERFORMANCE

Table 1: Model prediction performance. NN refers to an MLP for tabular and a ResNet for image datasets. The best-performing model for each metric is **bolded**, while the second-best is underlined.

| | Covid (Tabular) | | Diabetes (Tabular) | | Fund (Tabular) | | ISIC (Image) | | OCTMNIST (Image) | |
|---|---|---|---|---|---|---|---|---|---|---|
| | AUC (↑) | Acc (↑) | AUC (↑) | Acc (↑) | AUC (↑) | Acc (↑) | AUC (↑) | Acc (↑) | AUC (↑) | Acc (↑) |
| RN (Ours) | 0.95 ± 0.00 | 0.88 ± 0.00 | 0.97 ± 0.00 | 0.92 ± 0.01 | **0.75 ± 0.01** | **0.71 ± 0.00** | **0.91 ± 0.01** | **0.76 ± 0.01** | **0.99 ± 0.00** | **0.91 ± 0.02** |
| NN | 0.95 ± 0.00 | **0.89 ± 0.00** | 0.98 ± 0.00 | 0.93 ± 0.01 | 0.60 ± 0.02 | **0.71 ± 0.00** | 0.87 ± 0.01 | 0.70 ± 0.01 | **0.99 ± 0.01** | 0.88 ± 0.03 |
| MCD | 0.95 ± 0.00 | **0.89 ± 0.00** | 0.98 ± 0.00 | 0.93 ± 0.01 | 0.59 ± 0.02 | **0.71 ± 0.00** | 0.87 ± 0.01 | 0.70 ± 0.01 | **0.99 ± 0.01** | 0.88 ± 0.03 |
| DE | 0.95 ± 0.00 | **0.89 ± 0.00** | 0.98 ± 0.00 | 0.93 ± 0.00 | 0.59 ± 0.01 | **0.71 ± 0.00** | 0.89 ± 0.00 | 0.72 ± 0.01 | 0.99 ± 0.00 | 0.89 ± 0.02 |
| PN | **0.96 ± 0.01** | 0.88 ± 0.01 | 0.97 ± 0.00 | 0.92 ± 0.00 | 0.53 ± 0.00 | **0.71 ± 0.00** | 0.70 ± 0.01 | 0.64 ± 0.01 | 0.94 ± 0.02 | 0.80 ± 0.05 |
| BNN | 0.91 ± 0.02 | 0.85 ± 0.02 | 0.97 ± 0.01 | 0.91 ± 0.01 | 0.54 ± 0.01 | **0.71 ± 0.00** | 0.60 ± 0.02 | 0.52 ± 0.02 | **0.99 ± 0.00** | 0.87 ± 0.02 |
| EDL | 0.92 ± 0.02 | 0.79 ± 0.18 | 0.87 ± 0.07 | **0.95 ± 0.02** | 0.58 ± 0.05 | 0.62 ± 0.17 | 0.53 ± 0.01 | 0.51 ± 0.01 | 0.96 ± 0.00 | 0.83 ± 0.02 |

Table 1 compares model performance. ReconstructionNet (RN) achieved strong results across datasets, with second-highest AUC and accuracy on COVID and Diabetes, and the highest AUC and accuracy on Fund, ISIC, and OCTMNIST. Notably, on Fund, ReconstructionNet outperformed others in AUC despite similar accuracy, suggesting it learns discriminative features robust to data shifts (caused by the COVID-19 pandemic) and is resilient to epistemic uncertainty.

### 4.4.2 UNCERTAINTY ESTIMATION PERFORMANCE

**Aleatoric Uncertainty:** Table 2 summarizes uncertainty estimation performance. Reconstruction-Net outperformed all models on COVID except $\sigma$-risk at $\sigma = 0.1$, where it matched MLP Entropy and MCD. On Diabetes and Fund, it consistently ranked among the top across metrics.

**Distributional Uncertainty:** Table 3 presents the OOD detection performance of all uncertainty estimates. ReconstructionNet achieves the highest AUROC across all OOD datasets and improves as datasets deviate further from in-distribution data, showing it effectively ranks dataset dissimilarity. In contrast, other estimates like Entropy, MCD, PN, and BNN drop on highly dissimilar datasets (see ISIC) or struggle with unseen classes, as seen on OCTMNIST.

**Distinguishing Between Aleatoric and Distributional:** To illustrate ReconstructionNet's ability to separate uncertainty types, we visualize aleatoric and distributional uncertainty over time on the Fund dataset (Figure 2). Aleatoric uncertainty is highest before January 2020, while during the 2020

Table 2: Uncertainty estimation performance. The best-performing model for each metric is **bolded**, while the second-best model is underlined.

| Data | UE | Correlation (↑) | AURC (↓) | $\sigma = 0.1$ (↓) | $\sigma = 0.2$ (↓) | $\sigma = 0.3$ (↓) | $\sigma = 0.4$ (↓) |
|---|---|---|---|---|---|---|---|
| | RN (Ours) | **0.823 ± 0.016** | **0.028 ± 0.002** | 0.005 ± 0.001 | **0.006 ± 0.002** | **0.007 ± 0.002** | **0.012 ± 0.005** |
| Covid | MLP Entropy | 0.664 ± 0.012 | 0.030 ± 0.002 | **0.003 ± 0.004** | 0.016 ± 0.006 | 0.022 ± 0.006 | 0.032 ± 0.005 |
| | MLP MCD | 0.639 ± 0.012 | 0.032 ± 0.003 | 0.004 ± 0.005 | 0.018 ± 0.009 | 0.029 ± 0.006 | 0.042 ± 0.005 |
| | MLP DE | 0.603 ± 0.031 | 0.030 ± 0.003 | 0.011 ± 0.003 | 0.020 ± 0.007 | 0.032 ± 0.007 | 0.040 ± 0.005 |
| | PN Epis | 0.313 ± 0.082 | 0.038 ± 0.011 | 0.021 ± 0.010 | 0.034 ± 0.015 | 0.041 ± 0.017 | 0.051 ± 0.016 |
| | PN Alea | 0.690 ± 0.034 | 0.029 ± 0.007 | 0.016 ± 0.005 | 0.022 ± 0.007 | 0.032 ± 0.006 | 0.039 ± 0.005 |
| | BNN | 0.030 ± 0.027 | 0.132 ± 0.018 | 0.104 ± 0.070 | 0.159 ± 0.023 | 0.161 ± 0.023 | 0.132 ± 0.015 |
| | EDL | 0.674 ± 0.068 | 0.115 ± 0.094 | 0.238 ± 0.105 | 0.073 ± 0.009 | 0.058 ± 0.004 | 0.045 ± 0.005 |
| | RN (Ours) | 0.902 ± 0.011 | 0.008 ± 0.001 | **0.000 ± 0.000** | 0.001 ± 0.000 | **0.001 ± 0.000** | **0.001 ± 0.000** |
| Diabetes | MLP Entropy | 0.864 ± 0.009 | **0.007 ± 0.002** | **0.000 ± 0.000** | **0.000 ± 0.000** | **0.001 ± 0.000** | **0.001 ± 0.000** |
| | MLP MCD | 0.845 ± 0.022 | 0.008 ± 0.002 | **0.000 ± 0.000** | **0.000 ± 0.000** | **0.001 ± 0.000** | 0.002 ± 0.001 |
| | MLP DE | 0.733 ± 0.031 | 0.011 ± 0.001 | **0.000 ± 0.000** | 0.001 ± 0.000 | 0.005 ± 0.001 | 0.011 ± 0.003 |
| | PN Epis | 0.525 ± 0.297 | 0.020 ± 0.021 | 0.006 ± 0.009 | 0.016 ± 0.023 | 0.023 ± 0.030 | 0.028 ± 0.033 |
| | PN Alea | 0.812 ± 0.022 | 0.009 ± 0.001 | 0.001 ± 0.000 | 0.001 ± 0.000 | 0.002 ± 0.001 | 0.003 ± 0.001 |
| | BNN | -0.619 ± 0.012 | 0.214 ± 0.016 | 0.262 ± 0.024 | 0.222 ± 0.016 | 0.185 ± 0.016 | 0.136 ± 0.013 |
| | EDL | **0.942 ± 0.017** | 0.046 ± 0.034 | **0.000 ± 0.000** | 0.001 ± 0.002 | 0.006 ± 0.006 | 0.011 ± 0.010 |
| | RN (Ours) | **0.385 ± 0.008** | **0.141 ± 0.003** | 0.039 ± 0.004 | **0.048 ± 0.005** | **0.077 ± 0.015** | **0.126 ± 0.023** |
| Fund | MLP Entropy | 0.131 ± 0.034 | 0.212 ± 0.023 | 0.064 ± 0.043 | 0.132 ± 0.063 | 0.187 ± 0.036 | 0.241 ± 0.012 |
| | MLP MCD | 0.128 ± 0.038 | 0.236 ± 0.018 | 0.073 ± 0.067 | 0.124 ± 0.062 | 0.165 ± 0.046 | 0.210 ± 0.028 |
| | MLP DE | 0.022 ± 0.045 | 0.287 ± 0.032 | 0.364 ± 0.331 | 0.330 ± 0.192 | 0.283 ± 0.031 | 0.286 ± 0.045 |
| | PN Epis | 0.016 ± 0.001 | 0.291 ± 0.017 | 0.279 ± 0.028 | 0.278 ± 0.025 | 0.278 ± 0.022 | 0.279 ± 0.021 |
| | PN Alea | 0.166 ± 0.026 | 0.271 ± 0.001 | 0.221 ± 0.007 | 0.225 ± 0.009 | 0.230 ± 0.013 | 0.235 ± 0.009 |
| | BNN | -0.198 ± 0.028 | 0.364 ± 0.010 | 0.665 ± 0.189 | 0.543 ± 0.125 | 0.442 ± 0.065 | 0.405 ± 0.026 |
| | EDL | 0.162 ± 0.118 | 0.301 ± 0.202 | **0.016 ± 0.015** | 0.055 ± 0.062 | 0.101 ± 0.078 | 0.174 ± 0.092 |

Table 3: Out-of-distribution (OOD) detection performance, measured using AUROC. OOD datasets are presented in order of increasing deviation from the in-distribution. The best-performing model for each metric is **bolded**, while the second-best model is underlined.

| | ISIC | | | OCTMNIST | | |
|---|---|---|---|---|---|---|
| | BCN-IN (↑) | BCN-OUT (↑) | ChestMNIST (↑) | OCTDL-IN (↑) | OCTDL-OUT (↑) | ChestMNIST (↑) |
| RN (Ours) | **0.777 ± 0.086** | **0.846 ± 0.070** | **0.919 ± 0.097** | **0.783 ± 0.082** | **0.866 ± 0.049** | **1.000 ± 0.000** |
| ResNet Entropy | 0.742 ± 0.010 | 0.757 ± 0.027 | 0.664 ± 0.072 | 0.674 ± 0.031 | 0.386 ± 0.066 | 0.826 ± 0.060 |
| ResNet MCD | 0.746 ± 0.004 | 0.770 ± 0.016 | 0.678 ± 0.080 | 0.661 ± 0.034 | 0.392 ± 0.064 | 0.844 ± 0.060 |
| ResNet DE | 0.720 ± 0.020 | 0.703 ± 0.029 | 0.728 ± 0.055 | 0.739 ± 0.009 | 0.497 ± 0.029 | 0.900 ± 0.028 |
| PN Epis | 0.674 ± 0.021 | 0.686 ± 0.039 | 0.654 ± 0.050 | 0.565 ± 0.052 | 0.483 ± 0.087 | 0.470 ± 0.145 |
| PN Alea | 0.660 ± 0.026 | 0.669 ± 0.041 | 0.631 ± 0.065 | 0.625 ± 0.063 | 0.492 ± 0.025 | 0.500 ± 0.107 |
| BNN | 0.556 ± 0.011 | 0.540 ± 0.056 | 0.547 ± 0.059 | 0.742 ± 0.014 | 0.734 ± 0.016 | 0.896 ± 0.015 |
| EDL | 0.533 ± 0.011 | 0.539 ± 0.023 | 0.619 ± 0.024 | 0.633 ± 0.012 | 0.669 ± 0.011 | 0.760 ± 0.013 |

recession (per the National Bureau of Economic Research), distributional uncertainty predominates, demonstrating ReconstructionNet's discriminative capability.

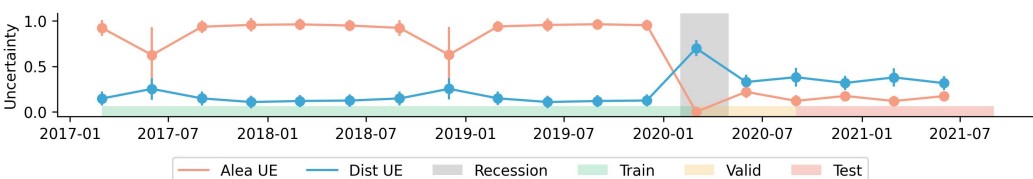

Figure 2: Aleatoric (Alea) and distributional (Dist) uncertainty over time on the Fund dataset. Uncertainty values are min–max normalized, and error bars represent one standard deviation. In early 2020, distributional uncertainty overtook aleatoric uncertainty due to COVID-19–induced shifts, while aleatoric uncertainty dropped as increasing class imbalance reduced label ambiguity.



(a) Label: AKIEC. Pred: BCC.  (b) Label: AKIEC. Pred: BKL.  (c) Label: BCC. Pred: BKL.

Figure 3: Uncertainty explanation illustration using images from the ISIC dataset. Positive attributions were min–max normalized, thresholded (pixel uncertainty > 0.15 shown in green), and overlaid for clarity. The highlighted regions "explain" the prediction uncertainty. AKIEC refers to Actinic Keratosis, BCC to Basal Cell Carcinoma, and BKL to Benign Keratosis.

### 4.4.3 UNCERTAINTY EXPLANATION PERFORMANCE

To highlight the practical value of our uncertainty explanations in identifying regions that contribute to misclassification, we qualitatively assess them on the ISIC dataset. In Figure 3a, the image was misclassified as BCC instead of AKIEC, with uncertainty concentrated on the mole at the top-left corner, likely because such dark, mole-like spots resemble features more typical of BCC than AKIEC (Lee, 2017). In Figures 3b and 3c, both images were misclassified as BKL, with uncertainty concentrated on the hairs covering a substantial portion of the lesion, as such occlusions can obscure diagnostic features. These visual explanations align with human intuition, demonstrating their effectiveness in pinpointing input features that confuse the model.

Table 4: Top-k accuracy of each uncertainty-explanation method in identifying perturbed features on the Covid and Diabetes datasets.

| Method | Covid | | | Diabetes | | |
|---|---|---|---|---|---|---|
| | Top-1 Acc | Top-3 Acc | Top-5 Acc | Top-1 Acc | Top-3 Acc | Top-5 Acc |
| IG | $0.163 \pm 0.003$ | $0.355 \pm 0.017$ | $0.436 \pm 0.020$ | $0.065 \pm 0.008$ | $0.221 \pm 0.005$ | $0.446 \pm 0.022$ |
| SHAP | $0.102 \pm 0.011$ | $0.193 \pm 0.020$ | $0.262 \pm 0.017$ | $0.093 \pm 0.004$ | $0.247 \pm 0.008$ | $0.387 \pm 0.014$ |
| RN (Ours) | $\mathbf{0.437 \pm 0.035}$ | $\mathbf{0.604 \pm 0.027}$ | $\mathbf{0.646 \pm 0.023}$ | $\mathbf{0.272 \pm 0.064}$ | $\mathbf{0.509 \pm 0.045}$ | $\mathbf{0.599 \pm 0.040}$ |

We also evaluate the performance of our uncertainty explanations on tabular data using the covariate-shift experiment from Watson et al. (2023). For each test instance, we randomly perturb one feature by adding noise drawn from $\mathcal{N}(0.5, 0.1)$ and assess whether an explanation ranks this perturbed feature among its top-k most uncertain features (Top-k Accuracy). This directly measures whether the explanation identifies the feature contributing to distributional uncertainty.

We compare ReconstructionNet (RN) explanations with: (1) Integrated Gradients (IG), a gradient-based method, and (2) KernelSHAP, a perturbation-based method, both applied to explain the entropy of ReconstructionNet's predictive distribution (Iversen et al., 2024). Across the COVID and Diabetes datasets (Table 4), RN achieves consistently higher top-k accuracy for all $k$, demonstrating its effectiveness in localizing the source of uncertainty.

## 5 CONCLUSION

This paper proposed ReconstructionNet, a neural network for reliable uncertainty estimation alongside classification. Unlike models based on conditional probability, ReconstructionNet uses class-specific autoencoders to model the input–output joint distribution, predicting the class with maximal joint probability (or minimal reconstruction error). This approach quantifies aleatoric and distributional uncertainty while minimizing epistemic uncertainty in a single training session. Across five real-world datasets, ReconstructionNet achieved comparable or improved classification performance, with uncertainty estimates showing superior reliability, selectivity, robustness to false negatives, and strong OOD detection. Its inbuilt, cost-free explanations highlight input features contributing to uncertainty, with theoretical properties of (1) Implementation Invariance, (2) Sensitivity, and (3) Consistency. While ReconstructionNet performs best on large, balanced datasets, this limitation suggests future directions, such as quantifying epistemic uncertainty.

## 6 REPRODUCIBILITY STATEMENT

For implementation details, see the code repository: `https://anonymous.4open.science/r/ReconstructionNet-4F8C/`. For tabular datasets, input features were normalized using min–max scaling prior to training. For image datasets, pixel intensities were scaled from [0, 255] to [0, 1].

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

# A APPENDIX

## A.1 DATASET DETAILS

### A.1.1 COVID-19 VIRUS TRANSMISSION DATASET

(Hinns et al., 2021) is a tabular binary classification dataset comprising 3,553 instances. Each instance is characterized by 32 continuous features describing the United Kingdom's COVID-19 poli-

cies and daily case counts across 12 regions, covering the period from February 2020 to February 2021. The dataset is labelled with a binary class indicating whether $R_t > 1$, where $R_t$ represents the effective reproduction number. A value of $R_t > 1$ signifies an increasing spread of COVID-19. We randomly divided the dataset into three subsets: training (70%), validation (10%), and test (20%).

### A.1.2 DIABETES DIAGNOSIS DATASET

(Mustafa, 2023) is a tabular binary classification dataset comprising 100,000 instances, each representing a patient. Each instance is described by eight features detailing the patient's demographics (e.g., age and gender), pre-existing conditions (e.g., prevalence of heart disease), and vital signs (e.g., blood glucose level), and has a binary label indicating whether the patient suffers from diabetes (0 indicating the patient is diabetes-free and 1 indicating the patient suffers from diabetes). We divided the dataset into three subsets for our experiments: training (80%), validation (10%), and test (10%).

### A.1.3 FUND PERFORMANCE EVALUATION DATASET

(Kovvuri et al., 2023) is a tabular binary classification dataset comprising 77,940 instances, designed to predict whether a fund's net asset value (NAV) exceeds its NAV from the previous quarter. Each instance represents the state of one of 4,330 funds between March 2017 and June 2021, sampled quarterly. Each instance is characterized by 18 continuous features, including macroeconomic indicators (such as stock market returns (ST), exchange rate returns (EXR), and interest rates (IR)), country-level equity investment percentages and net asset data for the fund (covering 11 countries and "Other Country"), the Herfindahl–Hirschman index (HHI), and a past performance metric computed as the sum of class labels from the past four quarters ("L4f Gain"). We split the dataset into three subsets based on date: instances before March 2020 were assigned to the training set (66.7%), instances between March 2020 and June 2020 to the validation set (11.1%), and instances after June 2020 to the test set (22.2%).

### A.1.4 ISIC

(Codella et al., 2019; Tschandl et al., 2018) is a dermoscopic image dataset containing instances from seven skin conditions: melanoma (MEL), melanocytic nevus (NV), basal cell carcinoma (BCC), actinic keratosis (AKIEC), benign keratosis (BKL), dermatofibroma (DF), and vascular lesion (VASC). We followed the 2018 ISIC Challenge split, with 10,015 training, 193 validation, and 1,512 test images. For the Out-of-Distribution (OOD) detection experiments, we constructed three OOD datasets with decreasing similarity to ISIC: (1) BCN-IN: 1,512 dermoscopic images of seen classes from the BCN20000 dataset (Combalia et al., 2019), (2) BCN-OUT: 313 images from the unseen class Scar, and (3) ChestMNIST: 1,512 images from the ChestMNIST dataset (Wang et al., 2017).

### A.1.5 OCTMNIST

(Kermany et al., 2018) is a retinal OCT image dataset comprising four classes: 47% Normal, 34% CNV (Choroidal Neovascularization), 11% DME (Diabetic Macular Edema), and 8% Drusen. We split the dataset into three subsets: 97,477 training, 10,832 validation, and 1,000 test instances. For the OOD detection experiments, we constructed three OOD datasets with decreasing similarity to OCTMNIST: (1) **OCTDL-IN**: 618 OCTDL (Kulyabin et al., 2024) images from seen classes, (2) **OCTDL-OUT**: 1,000 OCTDL images from unseen classes, and (3) **ChestMNIST**: 1,512 images from the ChestMNIST dataset (Wang et al., 2017).

### A.2 BASELINE DETAILS

We compared ReconstructionNet with recent state-of-the-art uncertainty estimates.

1. **Entropy** (Shannon, 1948) is derived from prediction probability.
2. **Monte Carlo Dropout (MCD)** (Gal & Ghahramani, 2016) We apply dropout ($p = 0.5$) to the penultimate layer of the model and keep dropout active during inference to yield $T$ predictions (where $T = 100$ for the Covid and Fund datasets, and $T = 10$ for all other datasets), following the hyperparameters in (Gal & Ghahramani, 2016).

3. **Deep Ensemble (DE)** (Lakshminarayanan et al., 2017) uses five models, with the standard deviation of their predictions as the uncertainty estimate.

4. **Posterior Network (PN)** (Charpentier et al., 2020) computes aleatoric uncertainty as the inverse of the maximum prediction probability and epistemic uncertainty as the inverse of the maximum of the Dirichlet distribution parameters.

5. **Bayesian Neural Networks (BNN)** (Jospin et al., 2022) estimate uncertainty as the standard deviation of $T = 100$ predictions obtained with different weight samples.

6. **Evidential Deep Learning (EDL)** (Sensoy et al., 2018) estimates uncertainty using the entropy of the predicted probabilities.

### A.3 METRIC DETAILS

#### A.3.1 CORRELATION

(Mi et al., 2022; Upadhyay et al., 2022) measures the reliability of the uncertainty estimate as the Pearson's correlation coefficient between the uncertainty estimate and the prediction error (measured as the absolute difference between the one-hot label and the predicted probability). A higher correlation indicates a more reliable estimate, as prediction errors are likely to be high when the model's uncertainty is high, and vice versa.

#### A.3.2 AURC (AREA UNDER RISK-COVERAGE CURVE)

(Ding et al., 2020) quantifies the selectivity of the uncertainty estimate, indicating its usefulness in selective prediction i.e. making predictions only for confident instances. This is computed as the area under the risk-coverage curve, which plots the 0/1 loss (Risk) for instances with uncertainty scores within the $\alpha$%-percentile (Coverage) against the coverage. A selective uncertainty estimate would yield a low AURC, as it effectively reduces loss across all possible uncertainty thresholds.

#### A.3.3 SIGMA-RISK SCORE

Evaluates the resilience of the uncertainty estimate to false negatives (incorrect instances with low uncertainty), which can lead to significant costs if many are left undetected. It is computed as the 0/1 loss of instances with normalized uncertainty less than $\sigma = \{0.1, 0.2, 0.3, 0.4\}$. A lower $\sigma$-risk score indicates greater robustness to false negatives. To ensure robustness to outliers, we apply min-max normalization with the minimum and maximum values computed after outlier removal with the interquartile range method.

#### A.3.4 OOD DETECTION

(Lakshminarayanan et al., 2017; Postels et al., 2020; Malinin & Gales, 2018) Evaluates how effectively the uncertainty estimate can distinguish between in-distribution and OOD instances. This is quantified using AUROC (Area Under the Receiver Operating Characteristic Curve), where the true label indicates whether an instance is OOD and the target score is the uncertainty estimate. A higher AUROC reflects a more reliable uncertainty measure capable of identifying OOD inputs.

### A.4 IMPLEMENTATION DETAILS

The use of error weights $\omega$ allows the model to decouple the modeling of joint prediction probabilities from the classification task, enabling simultaneous optimization of both objectives. It also dynamically scales errors across classes to address variations in reconstruction difficulty and adjusts the contribution of each feature to the prediction, acknowledging that not all reconstruction errors equally impact the final outcome.

For image datasets, to reduce the size of the ReconstructionNet model, we trained class-specific decoders sharing a common encoder instead of training separate class-specific autoencoders. The encoder was based on a ResNet18 (He et al., 2015) backbone, and the decoders were implemented as inverted ResNet18 backbones, replacing convolutional layers with transposed convolutions.

Hyperparameters for all models were determined using grid search on the validation set. For ReconstructionNet, we tuned the compression ratio (which determines the latent vector length as its product with the feature count) along with the number of encoder and decoder layers, the width of intermediate layers, and the loss function parameter $\beta$ ($\beta = 1.25, 2, 0.75$ for Covid, Diabetes, and Fund datasets). Both the MLP and ReconstructionNet models were trained on the training set with oversampling using SMOTE (Chawla et al., 2002). All models were trained using the Adam optimizer with early stopping. Each experiment was repeated five times, and we report the mean and standard deviation of each metric.

## A.5 VERIFYING UNCERTAINTY EXPLANATION PROPERTIES WITH TOY EXAMPLES

### A.5.1 IMPLEMENTATION INVARIANCE

We illustrate the property of implementation invariance with a simple example. Suppose we have two ReconstructionNet models, $f$ and $f'$, that differ in architecture:

- $f$: Each class autoencoder is shallow and linear;
- $f'$: Each class autoencoder is deeper with nonlinearities.

If for every input $\mathbf{x}$, the two ReconstructionNet models yield identical prediction probabilities $\hat{p}$ for all classes, then:

1. The predicted classes $c^* = \arg\max_j \hat{p}_j$ are identical,
2. The weighted reconstruction errors $\epsilon$ are identical,
3. The uncertainty explanations $\zeta(\mathbf{x})$, which are the feature-wise weighted reconstruction errors of the predicted classes, are identical.

Thus, ReconstructionNet's uncertainty explanations are *implementation-invariant* under this definition of functional equivalence.

### A.5.2 SENSITIVITY

We illustrate the sensitivity property with a simple example. Consider a dataset where each input $\mathbf{x}$ has three features, $(x^1, x^2, x^3)$. Suppose that for the predicted class $c^*$:

- The weighted reconstruction errors depend only on $x^1$ and $x^2$;
- The weight for $x^3$ is zero: $w^3_{c^*} = 0$, indicating that changes in $x^3$ do not affect the class prediction.

The uncertainty explanation for input $\mathbf{x}$ is given by:

$$\zeta(\mathbf{x}) = \begin{bmatrix} w^1_{c^*}\|x^1 - \hat{x}^1_{c^*}\|^2 & w^2_{c^*}\|x^2 - \hat{x}^2_{c^*}\|^2 & w^3_{c^*}\|x^3 - \hat{x}^3_{c^*}\|^2 \end{bmatrix} \tag{15}$$

Since $w^3_{c^*} = 0$, the explanation assigns zero attribution to $x^3$, which is irrelevant to the prediction. Hence, ReconstructionNet's uncertainty explanations satisfy the *sensitivity* property: features irrelevant to the prediction receive zero attribution in the weighted reconstruction error.

### A.5.3 CONSISTENCY

We illustrate the consistency property with a simple example. Consider an input $\mathbf{x} = (x^1, x^2, x^3)$; For the predicted class $c^*$, the uncertainty explanation is:

$$\zeta(\mathbf{x}) = \begin{bmatrix} w^1_{c^*}\|x^1 - \hat{x}^1_{c^*}\|^2 & w^2_{c^*}\|x^2 - \hat{x}^2_{c^*}\|^2 & w^3_{c^*}\|x^3 - \hat{x}^3_{c^*}\|^2 \end{bmatrix} \tag{16}$$

Now consider a modification $f'$ of the model $f$ where feature $x^2$ is made more important for the prediction (i.e., its weight increases in the softmax over reconstruction errors).

- The reconstruction errors $\|x^j - \hat{x}^j_{c^*}\|^2$ remain unchanged, since the autoencoders are unchanged.
- The weight for $x^2$ increases: $w^{2'}_{c^*} > w^2_{c^*}$.

Then the uncertainty attribution for $x_2$ becomes:

$$\zeta(\mathbf{x})^{2'} = w_{c^*}^{2'} \|x_2 - \hat{x}_{c^*}^2\|^2 \geq w_{c^*}^2 \|x^2 - \hat{x}_{c^*}^2\|^2 = \zeta(\mathbf{x})^2. \tag{17}$$

Thus, increasing a feature's contribution to the prediction does not decrease its uncertainty attribution, demonstrating that ReconstructionNet's uncertainty explanations satisfy the *consistency* property.

### A.6 ADDITIONAL UNCERTAINTY EXPLANATION EXAMPLES

**MNIST:** The examples illustrate how the model's uncertainty explanations correspond to ambiguous digit features that resemble other classes. In Figure 4a, the model predicted a 2 instead of a 0, showing uncertainty around the missing connection in the 0. In Figure 4b, it predicted a 6 instead of a 0, with uncertainty focused on the tail of the 6. In Figure 4c, the model again predicted a 6 instead of a 0, highlighting uncertainty on the right curve that resembles a 0. In Figure 4d, it predicted a 9 instead of a 4, with uncertainty at the connecting bump of the 4 (which is atypical in a 9). In Figure 4e, the model predicted a 5, with uncertainty concentrated on the top tail of the digit. Finally, in Figure 4f, it predicted a 3 instead of an 8, highlighting the lower-left connecting curve, whose removal would make the digit resemble a 3 more strongly.

**ISIC:** The examples demonstrate how the model's uncertainty explanations capture visual features that increase ambiguity in lesion classification, leading to misclassifications. In Figures 4g and 4h, the model misdiagnoses the dermoscopic image, with the uncertainty explanations highlighting a scab-like region. In Figures 4i, 4j and 4k, images were misclassified as MEL (melanoma) instead of BCC or AKIEC, with uncertainty concentrated on the uneven pigmentation of the lesion, a feature often associated with melanoma. In Figure 4l, the image was misclassified as AKIEC with uncertainty explanations highlighting the presence of hairs covering a substantial portion of the lesion.

**OCTMNIST** These cases illustrate how the model's uncertainty highlights regions that contribute to misclassification in OCT images. In Figure 4m, the model classified the OCT image as DME, with uncertainty concentrated around one of the fluid pockets. In Figure 4n, the model predicted CNV instead of DME, again highlighting a fluid pocket. This reflects the ambiguity introduced by fluid pockets, which are common to both conditions. Figures 4o, 4p, 4q, and 4r are Drusen images misclassified as other conditions, with uncertainty explanations highlighting the deposits (uneven bumps in the OCT images), which are key features of Drusen and major contributors to the model's uncertainty in each incorrect prediction.

### A.7 COMPARISON WITH LLMS

Figure 5 compares uncertainty explanations from ReconstructionNet and a publicly available, off-the-shelf large language model (LLM; Microsoft Copilot). We observe that ReconstructionNet produces more selective explanations, highlighting only small regions that contribute to uncertainty, whereas Copilot often highlights large areas (notably, in Figure 5b the entire digit is highlighted). Moreover, the regions identified by ReconstructionNet are more accurate. For example, in Figure 5a, where the digit 4 is misclassified as 7, ReconstructionNet correctly highlights the additional vertical line on the right, while Copilot highlights the connecting region that is common to both 4 and 7 and should not confuse the model. Similarly, in Figure 5c, where a 6 is misclassified as 5, ReconstructionNet pinpoints the extra vertical line on the bottom left that distinguishes 5 from 6, whereas Copilot highlights the top-left vertical line, which can occur in both digits.

### A.8 RELATION TO CONTRASTIVE LEARNING AND SIAMESE NETWORKS

Note that, although the training procedure of ReconstructionNet shares similarities with contrastive learning (Chopra et al., 2005), as both aim to minimize the distance between instances of the same class while maximizing the distance between instances of different classes, there are key differences in how these distances are computed. In contrastive learning, the distance is explicitly calculated as the L2-norm between the embeddings of a pair of instances. In ReconstructionNet, the distance between an instance and the instances of a particular class is represented as the reconstruction error of the corresponding class autoencoder. This distinction in distance measurement leads to differences in the training loss. While contrastive learning incorporates the L2-norm distance between pairs of instances, ReconstructionNet uses class reconstruction error for its loss function.

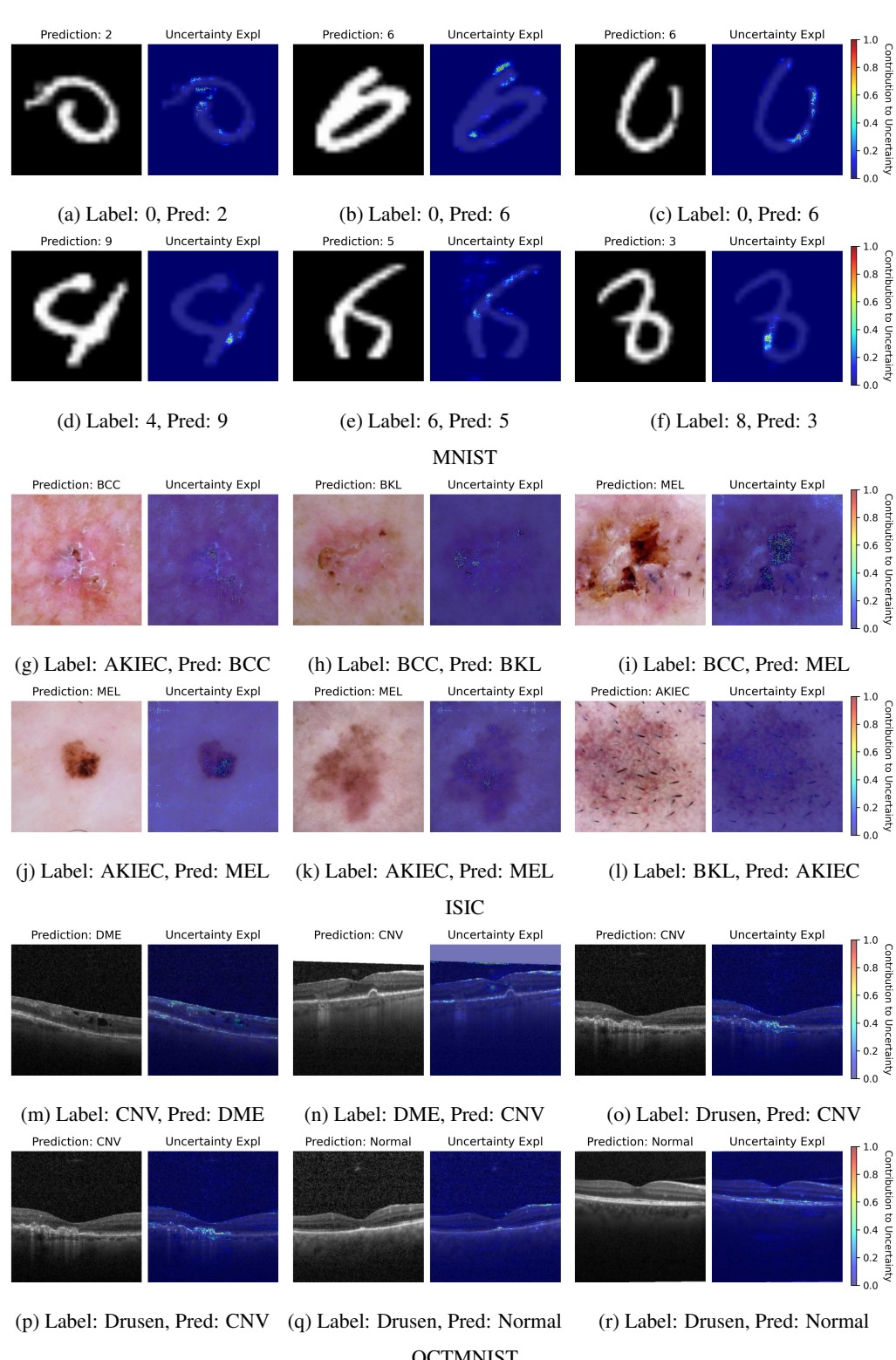

(a) Label: 0, Pred: 2    (b) Label: 0, Pred: 6    (c) Label: 0, Pred: 6

(d) Label: 4, Pred: 9    (e) Label: 6, Pred: 5    (f) Label: 8, Pred: 3

MNIST

(g) Label: AKIEC, Pred: BCC    (h) Label: BCC, Pred: BKL    (i) Label: BCC, Pred: MEL

(j) Label: AKIEC, Pred: MEL    (k) Label: AKIEC, Pred: MEL    (l) Label: BKL, Pred: AKIEC

ISIC

(m) Label: CNV, Pred: DME    (n) Label: DME, Pred: CNV    (o) Label: Drusen, Pred: CNV

(p) Label: Drusen, Pred: CNV    (q) Label: Drusen, Pred: Normal    (r) Label: Drusen, Pred: Normal

OCTMNIST

Figure 4: Uncertainty explanation illustration for MNIST, ISIC and OCTMNIST datasets. Positive uncertainty explanations were min-max normalized, gamma-corrected, and overlaid on the images for clarity. The highlighted regions "explain" the prediction uncertainty.

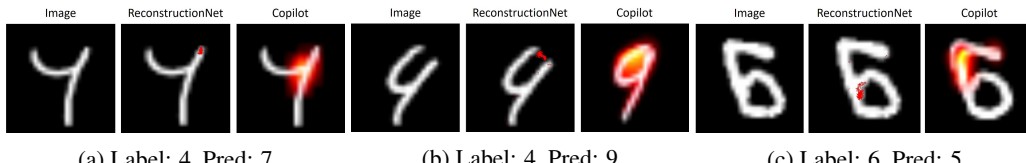

(a) Label: 4, Pred: 7          (b) Label: 4, Pred: 9          (c) Label: 6, Pred: 5

Figure 5: Comparison of uncertainty explanations from ReconstructionNet and Microsoft Copilot on the MNIST dataset. Positive uncertainty explanations from ReconstructionNet were min–max normalized, with pixels having attribution greater than 0.2 shown in red and overlaid on the images for clarity. Microsoft Copilot explanations were generated using the prompt: "This image has been misclassified as [image class prediction]. Generate an image highlighting the regions contributing to prediction uncertainty in red." The highlighted regions are intended to "explain" the prediction uncertainty.

Furthermore, in contrastive learning, the distances between instances of different classes are explicitly maximized and included in the loss function. In contrast, ReconstructionNet exclusively trains each class autoencoder with instances belonging to its specific class. The architecture of ReconstructionNet shares similarities with Siamese networks (Bromley et al., 1993) in its use of sub-networks. However, unlike Siamese networks, which share weights across sub-networks, ReconstructionNet allows each class autoencoder to have distinct weights.

