# OpenReview forum: "ReconstructionNet: A Neural Network Architecture for Uncertainty-Aware Predictions with Explainability"
_ICLR.cc/2026/Conference — Submitted to ICLR 2026_

### Official Review · Reviewer_vGHo · 2025-10-28

**Soundness:** 2
**Presentation:** 3
**Contribution:** 2
**Rating:** 4
**Confidence:** 2

**Summary:**

The paper proposes ReconstructionNet, which models the input-output joint distribution using class-specific autoencoders. In a single forward pass, the model simultaneously provides predictions, distributional (reconstruction error) uncertainty, and aleatoric uncertainty (entropy of the prediction). The reported classification performance on multiple real-world datasets (tabular and medical images) is comparable to or better than baselines.

**Strengths:**

The proposed uncertainty estimation method does not require additional multiple sampling or other modules, and it distinguishes between aleatoric and distributional uncertainty. The paper also provides relatively thorough theoretical proofs.

**Weaknesses:**

1. The related work on Bayesian Methods in Section 1.2.1 appears somewhat outdated. The statement that they are "usually computationally expensive" is somewhat one-sided. In recent years, many studies on Bayesian uncertainty estimation have also been computationally efficient, such as Bayesian Last Layer-type methods ([1] and [2]), which reduce the computational overhead of forward propagation by restricting stochasticity to specific layers.

2. In Section 3.2, the authors use θ₁ and θ₂ to distinguish between the two types of uncertainty in the joint probability formulation. However, this seems to be used only for conceptual characterization and does not appear to be involved in the actual network optimization or uncertainty evaluation process.

3. Using reconstruction error as uncertainty seems to heavily rely on the idea of reconstruction-based Out-of-Distribution (OOD) detection. The assumption inherent in the network architecture is that samples within a class are considered in-distribution, while samples from different classes are treated as out-of-distribution.


[1] Harrison, James, John Willes, and Jasper Snoek. "Variational Bayesian Last Layers." Fifth Symposium on Advances in Approximate Bayesian Inference.


[2] Hu, Xinyue, et al. "Enhancing Uncertainty Estimation and Interpretability with Bayesian Non-negative Decision Layer." The Thirteenth International Conference on Learning Representations.

**Questions:**

1.   What is the rationale behind treating the poorly reconstructed part (i.e., high reconstruction error) as having high distributional uncertainty? How is it ensured that this model is well-calibrated? It is recommended to supplement the experiments with metrics such as Expected Calibration Error (ECE) and Negative Log-Likelihood (NLL).

2. A major concern arises from the training cost: there is one autoencoder per class. How is the training overhead managed? How scalable is the method for large-scale datasets with a large number of classes (e.g., ImageNet-1K)?

---

> ### Author Response · Authors · 2025-11-19
>
> Thank you for your constructive and insightful feedback. Our responses are provided below:
>
> 1.  **“Related work...”**
>
>     We have updated our related work section to include recent efficient Bayesian methods, such as Bayesian Last Layer approaches, clarifying that computational cost can be reduced in these models.
>
> 2.  **“In Section 3.2...”**
>
>     Section 3.2 is intended to conceptually illustrate how our method captures the two types of uncertainty. The thresholds θ₁ and θ₂ are used to mathematically characterize aleatoric and distributional uncertainty for clarity and explanation, but they are not directly involved in network optimization or the computation of our uncertainty estimates.
>
>     The section first defines the two types of uncertainty formally and shows how they can be quantified from the joint probabilities. In the subsequent paragraphs, we link these definitions to our actual uncertainty measures (Shannon entropy for aleatoric uncertainty and reconstruction error for distributional uncertainty). This structure ensures a clear connection between the formal definitions and the practical estimates produced by ReconstructionNet.
>
> 3.  **“Using reconstruction error as uncertainty...”**
>
>     While ReconstructionNet leverages reconstruction error, it differs fundamentally from traditional reconstruction-based OOD detection:
>
>     1. *Joint input–output distribution for classification:* ReconstructionNet uses class-specific autoencoders to model the joint distribution of inputs and outputs for each class. Classification is performed by selecting the class with the highest joint probability (or lowest reconstruction error relative to that class), rather than treating other classes as generic OOD.
>     2. *Uncertainty estimation:* Reconstruction error is used to quantify both aleatoric and distributional uncertainty, not just detect OOD samples.
>     3. *Interpretability:* Feature-level reconstruction errors provide explanations for uncertain predictions, offering interpretable insights beyond standard OOD detection.
>
> 4.  **“rationale behind treating the poorly reconstructed part…”**
>
>     In ReconstructionNet, each class-specific autoencoder models the joint input–output distribution for its class. The reconstruction error of each feature reflects how well the model has learned the typical patterns for that feature in the predicted class.
>
>     - Features with high reconstruction error indicate that the input deviates from what the model has seen during training for that class.
>     - By weighting these errors according to feature importance for the prediction, we obtain feature-level contributions to distributional uncertainty, ensuring that only deviations that matter for classification are emphasized.
>
>     Thus, high feature-wise reconstruction error corresponds to high distributional uncertainty because it identifies the features that make the input “out-of-distribution” relative to the predicted class.
>
> 5. **“well-calibrated”**
>
>     Our work focuses on uncertainty quantification, specifically producing uncertainty estimates that reliably order inputs from most confident to most uncertain, rather than directly addressing calibration. Metrics such as Expected Calibration Error (ECE) and Negative Log-Likelihood (NLL) evaluate calibration, which is distinct from our goal of capturing relative uncertainty.
>
>     For future work, existing calibration techniques, such as temperature scaling, could be applied on top of ReconstructionNet’s uncertainty estimates to produce well-calibrated uncertainties while preserving the reliable ordering provided by our method.
>
> 6. **“training cost…”**
>
>     We appreciate the reviewer’s comment regarding scalability. Our work focuses on domain-specific applications where the number of classes is typically much smaller than in ImageNet. For example, the dermoscopic image dataset contains seven skin conditions (melanoma, melanocytic nevus, basal cell carcinoma, actinic keratosis, benign keratosis, dermatofibroma, and vascular lesions), and the retinal OCT dataset contains four classes (Normal, CNV, DME, and Drusen).
>
>     In these applications, being able to quantify uncertainty, explain predictions, and reliably distinguish between classes is of high practical value. We acknowledge that the class-specific autoencoder architecture may face challenges as the number of classes grows substantially, and scaling to datasets with hundreds or thousands of classes would require additional considerations. Nonetheless, for the targeted domains of this study, the architecture provides meaningful and interpretable uncertainty estimates alongside accurate classification.

---

> > ### Comment · Reviewer_vGHo · 2025-11-26
> >
> > Thank you for the authors’ response. However, I still believe that this paper falls below the acceptance bar. The most critical issue (also acknowledged by the authors) is the lack of scalability: the method requires constructing a separate network for each class and performing inference over all $C$ classes, which poses a serious challenge when used with moderately large backbones. The authors also note that their approach is primarily intended for domain-specific applications with a small number of classes. However, even in such settings, there exist many more lightweight and scalable uncertainty estimation methods. As a result, the advantages of the proposed method are not sufficiently clear, and I therefore decide to maintain my score.

---

### Official Review · Reviewer_mVt2 · 2025-10-30

**Soundness:** 2
**Presentation:** 3
**Contribution:** 3
**Rating:** 4
**Confidence:** 3

**Summary:**

This paper focuses on the problems of uncertainty and the explanation in neural networks. Existing uncertainty estimation methods either capture only one type of uncertainty or require high computational cost, and they typically lack explanations regarding the source of uncertainty. To address these limitations, this paper proposes ReconstructionNet, a new architecture designed to predict and quantify different types of uncertainty while providing built-in interpretability. Experiments conducted on both tabular and image datasets demonstrate the effectiveness of the ReconstructionNet in terms of the prediction accuracy, reliable uncertainty estimates, and intuitive uncertainty explanations that highlight regions contributing to model uncertainty.

**Strengths:**

1. The proposed neural network architecture offers multiple desirable properties, including the ability to estimate both aleatoric and distributional uncertainty within a single model, while also providing inherent uncertainty explanations without the need for additional modules.
2. The paper includes a theoretical analysis of the proposed uncertainty explanations, discussing properties like the sensitivity and consistency.

**Weaknesses:**

1. The paper claims that the proposed architecture can capture different types of uncertainty and minimize epistemic uncertainty. However, epistemic uncertainty is only briefly mentioned in Table 1 as part of the prediction performance. There is no explicit evaluation, baseline comparisons, or in-depth analysis of its estimation performance.
2. The paper focuses solely on uncertainty estimation methods in the experiments, but does not include or discuss uncertainty calibration methods such as Temperature Scaling [1] or Parametrized Temperature Scaling [2].
3. The uncertainty evaluation mainly considers the correlation and AURC metrics. The other commonly used uncertainty error metrics, such as Expected Calibration Error (ECE) or Adaptive Calibration Error (ACE), are not included.
4. In terms of uncertainty explanation evaluation, the paper discusses gradient-based and perturbation-based methods in the Related Work section, but it does not provide any quantitative or qualitative comparisons with these methods to validate the overall performance of its explanations.

[1] On Calibration of Modern Neural Networks. ICML 2017.

[2] Parameterized Temperature Scaling for Boosting the Expressive Power in Post-Hoc Uncertainty Calibration. ECCV 2022.

**Questions:**

1. The paper only presents uncertainty explanations for image data. How does the proposed method perform on tabular data, and how much interpretable or supportive explanation information can be derived in that context?
2. The construction details of the proposed architecture are not clear. Is ReconstructionNet built upon existing neural network architectures, and can it be adapted or integrated with different model types?

---

> ### Author Response · Authors · 2025-11-19
>
> Thank you for your constructive and insightful feedback. Our responses are provided below:
>
> 1. **“epistemic uncertainty”**
>
>     In real-world problems, it is difficult to disentangle aleatoric, epistemic, and distributional uncertainty in a fully isolated manner, as ground-truth uncertainty values are fundamentally unobservable in real-world settings. The intuition behind our approach is not to estimate epistemic uncertainty in isolation, but to reduce it by constraining the hypothesis space of ReconstructionNet. As noted in the paper, “the architecture and loss function of ReconstructionNet limit its hypothesis space, making it more resistant to epistemic uncertainty.” Therefore, our evaluation focuses on downstream effects, namely improved robustness under distribution shifts (seen in Table 1).
>
> 2. **“uncertainty calibration...”**
>
>     Our work focuses on uncertainty quantification, producing estimates that reliably rank inputs from most confident to most uncertain, evaluated via correlation-based metrics and AURC. Metrics like ACE/ECE assess uncertainty calibration, a related but distinct goal. Since our primary aim is capturing relative uncertainty, calibration metrics were not included. For future work, post-hoc calibration methods such as Temperature Scaling could be applied to ReconstructionNet’s estimates to produce well-calibrated uncertainties while preserving the reliable ranking provided by our method.
>
> 3. **“uncertainty explanation evaluation…”**
>
>     We thank the reviewer for highlighting the need to evaluate uncertainty explanations beyond image data and to compare against existing approaches. We have added a new experiment to assess explanation quality on tabular data, adapting the covariate shift evaluation protocol from [1]. For each test instance, we randomly perturb one feature by adding noise drawn from N(0.5,0.1), and evaluate whether an uncertainty explanation correctly ranks the perturbed feature among its top-k most uncertain features (Top-k Accuracy). This directly measures whether explanations successfully identify the feature contributing to distributional uncertainty.
>
>     We compare ReconstructionNet (RN) explanations against:
>     1. Integrated Gradients (IG) – a gradient-based method [2], and
>     2. KernelSHAP – a perturbation-based method applied to explain the entropy of ReconstructionNet’s predictive distribution [2].
>
>     We conduct the evaluation on the COVID-19 and Diabetes tabular datasets over multiple runs. As shown below, ReconstructionNet yields consistently higher Top-k Accuracy across all k ∈ {1,3,5}, demonstrating its ability to correctly localize the source of uncertainty.
>
>     |||Covid|||Diabetes||
>     |-|-|-|-|-|-|-|
>     |Method|Top-1 Acc|Top-3 Acc|Top-5 Acc|Top-1 Acc|Top-3 Acc|Top-5 Acc|
>     |IG|0.163 ± 0.003|0.355 ± 0.017|0.436 ± 0.020|0.065 ± 0.008|0.221 ± 0.005|0.446 ± 0.022|
>     |SHAP|0.102 ± 0.011|0.193 ± 0.020|0.262 ± 0.017|0.093 ± 0.004|0.247 ± 0.008|0.387 ± 0.014|
>     |RN (Ours)|0.437 ± 0.035|0.604 ± 0.027|0.646 ± 0.023|0.272 ± 0.064|0.509 ± 0.045|0.599 ± 0.040|
>
>     These results validate that ReconstructionNet explanations not only perform well on images, but also provide interpretable and supportive explanations for tabular data. In this context, ReconstructionNet highlights features that confuse the model under covariate shift, helping practitioners understand which risk factors or clinical indicators are contributing most to distributional uncertainty. We have incorporated this experiment and discussion into the revised manuscript (see Section 4.4.3).
>
> 4. **“construction details…”**
>
>     We appreciate the reviewer’s comment. ReconstructionNet refers to the overall framework comprising:
>     1. class-specific autoencoders,
>     2. class-specific reconstruction error weights, and
>     3. a negated softmax aggregation to produce uncertainty-aware predictions.
>
>     Importantly, the architecture of the encoder and decoder networks is not fixed in our design. ReconstructionNet is intended to be a model-agnostic framework rather than a single neural architecture. This flexibility allows it to be adapted to different data modalities and backbone types.
>
>     To demonstrate this versatility, our experiments include:
>     - MLP-based autoencoders for tabular datasets, and
>     - ResNet-18-based autoencoders for image datasets, as detailed in Appendix Section A.4.
>
>     To further clarify implementation details, we have updated the caption of Figure 1 to include a pointer to Appendix A.4 to make the construction details clearer.
>
> **References:**
> 1. Watson, D., O'Hara, J., Tax, N., Mudd, R. and Guy, I., 2023. Explaining predictive uncertainty with information theoretic shapley values. Advances in Neural Information Processing Systems, 36, pp.7330-7350.
> 2. Iversen, P., Witzke, S., Baum, K. and Renard, B.Y., 2023. Identifying drivers of predictive aleatoric uncertainty. arXiv preprint arXiv:2312.07252.

---

### Official Review · Reviewer_u4aB · 2025-10-31

**Soundness:** 2
**Presentation:** 2
**Contribution:** 2
**Rating:** 4
**Confidence:** 3

**Summary:**

This paper proposes a novel uncertainty estimation method capable of modeling different types of uncertainty. This method relies on a class-specific autoencoder and computes uncertainty by calculating the discrepancy between samples and their reconstructions.

It is important to note that this method is similar to reconstruction-based anomaly detection methods; thus, a sufficient comparison with anomaly detection methods and an enumeration of their differences are necessary.

Additionally, the experiments of this model have been mainly conducted on simple datasets, and how to extend it to large-scale experiments also requires further clarification.

**Strengths:**

1. can model different types of uncertainty, including aleatoric, epistemic, and distributional uncertainty.

2. Can give an explanation for uncertainty.

3. Achieving good performance compared to baseline models.

**Weaknesses:**

1. The connection and difference between the proposed method and autoencoder-based anomaly detection methods.

2. How to scale to large-scale datasets.

**Questions:**

See weakness.

---

> ### Author Response · Authors · 2025-11-19
>
> Thank you for your constructive and insightful feedback. Our responses are provided below:
>
> 1.  **“The connection and difference...”**
> While our method uses reconstruction error, it differs substantially from conventional autoencoder-based anomaly detection.
>
>     1. *Role of reconstruction error.* Anomaly-detection autoencoders use reconstruction error only to detect deviations from the training distribution. In contrast, ReconstructionNet models the input-output joint distribution using class-specific autoencoders, and performs classification by selecting the class with the highest joint likelihood (or lowest reconstruction error). Thus, reconstruction is central to both classification and uncertainty estimation, not only distributional uncertainty estimation.
>     2. *Training objective and uncertainty types.* Unlike unsupervised losses used in anomaly detection, ReconstructionNet uses a supervised objective that yields both aleatoric and distributional uncertainty in a single training process, while minimising epistemic uncertainty.
>     3. *Explanations.* Reconstruction patterns provide feature-level uncertainty explanations, which differ from scalar anomaly scores typically produced by AE-based detectors.
>
>     We note that our Related Work section discusses reconstruction-based uncertainty methods, including the recent Reconstruction Uncertainty Estimate (RUE), and highlights how our approach extends this line of work by quantifying both aleatoric and distributional uncertainty within a single model. To address this comment directly, we have also updated the Related Work section with an additional subsection that explicitly discusses autoencoder-based anomaly detection methods and clarifies how ReconstructionNet differs from them.
>
> 2.  **“How to scale to large-scale datasets.”**
> The performance of ReconstructionNet is not affected by dataset size. All class-specific autoencoders are trained concurrently, so scaling of training time is comparable to a standard CNN.

---

### Official Review · Reviewer_QBBi · 2025-11-01

**Soundness:** 1
**Presentation:** 2
**Contribution:** 1
**Rating:** 2
**Confidence:** 5

**Summary:**

This paper proposes a reconstruction-guided image classification with uncertainty quantification. They compute aleatoric uncertainty using entropy-based calculations of the classification probabilities, and epistemic uncertainty using the reconstruction error. Their proposed method is evaluated with six different methods of UQ in six different datasets. Experimental results demonstrate the superior performance of their proposed method.

**Strengths:**

**1. Good literature review.** The paper is enriched with discussion of related works in uncertainty quantification. They have discussed Bayesian, Evidential, and Deterministic methods of uncertainty quantification.

**2. Strong experimental results.** This paper conducts extensive evaluation with different datasets and compares with different baselines. They report prediction performance, quality of estimated uncertainty, and OOD detection.

**Weaknesses:**

**1. Limited novelty:** The reconstruction loss as a regularizer for a image classification task is not novel and have been used in many other previous works [1,2]. A proper literature review of those works is required to better understand the novelty of this work.

**2. Poor writing:** The paper is filled with many definitions which are standard in ML and UQ literature. Those definitions are not the contribution of this paper and including them wastes writing resources.

**3. Confusing uncertainty decomposition.** Uncertainty in deep models are divided into aleatoric and epistemic uncertainty. The distribution uncertainty is a subset of epistemic uncertainty specific to domain-generalization. However, the paper interchangeably utilize these terms and it beccomes confusing to the reader.

**4. Scalability.** Another serious limitation of this paper is its architecture choice. It essentially creates $C$ encoder to perform the reconstruction and classification. However, as class number $C$ grows, this paper's method will struggle to generalise. This is an important limitation of this model's architecture choice. For example, this method is not applicable to any ImageNet classification system.

[1] Le, L., Patterson, A. and White, M., 2018. Supervised autoencoders: Improving generalization performance with unsupervised regularizers. Advances in neural information processing systems, 31.

[2] Ghifary, M., Kleijn, W.B., Zhang, M., Balduzzi, D. and Li, W., 2016, September. Deep reconstruction-classification networks for unsupervised domain adaptation. In European conference on computer vision (pp. 597-613). Cham: Springer International Publishing.

**Questions:**

N/A

---

> ### Author Response · Authors · 2025-11-19
>
> Thank you for your insightful feedback. Our responses are provided below:
> 1.  **“Limited novelty...”**
>
>     We thank the reviewer for the suggestion to contextualize our work with related reconstruction-based approaches. We have updated the manuscript to include discussion of both papers highlighting key distinctions.
>
>     While these works use reconstruction loss as a regularizer for classification, ReconstructionNet differs fundamentally:
>     1.  *Class-specific autoencoders and joint distribution modeling.* Unlike the cited works, which use a single autoencoder and rely on conditional probabilities for classification, ReconstructionNet employs one autoencoder per class to model the input-output joint input-output distribution. Classification is performed by selecting the class with the lowest reconstruction error, rather than using conditional probabilities.
>     2.  *Uncertainty estimation and explanations.* Prior works do not utilize reconstruction error to quantify aleatoric or distributional uncertainty, nor to provide feature-level explanations for model predictions. ReconstructionNet leverages reconstruction errors both for reliable uncertainty estimation and for interpretable insights into which input features drive uncertainty.
>     3.  *Distinct architecture and prediction mechanism.* In the prior works, encoders are directly connected to the prediction head. In contrast, ReconstructionNet computes weighted reconstruction errors across class-specific autoencoders to generate predictions, representing a fundamentally different approach to combining reconstruction and classification.
>
> 2.  **“Many definitions...”**
>
>     To help improve the manuscript, could the reviewer indicate which definitions they consider unnecessary, so we can streamline the text without losing clarity.
>
> 3.  **“Confusing uncertainty decomposition…”**
>
>     In our work, we follow the definitions of uncertainty used in prior literature that explicitly separate three types of uncertainty (as described in Section 2.1):
>     1. *Aleatoric uncertainty* arises from inherent noise in the training data, such as overlapping classes in classification tasks.
>     2. *Epistemic uncertainty* stems from lack of knowledge in model parameters, which can be reduced by increasing data or constraining the hypothesis space.
>     3. *Distributional uncertainty* captures uncertainty caused by shifts between training and prediction distributions.
>
>     These definitions are consistent with Malinin & Gales (2018) [1], Durasov et al. (2021) [2], and the recent survey by Gawlikowski et al. (2023) [3], which distinguish distributional uncertainty from general epistemic uncertainty to explicitly address domain shift and out-of-distribution scenarios.
>
>     To help us improve the manuscript, could the reviewer kindly indicate the specific sections where the terms are perceived to be used interchangeably? We have aimed to use each uncertainty type consistently according to the definitions in Section 2.1.
>
> 4.  **“Scalability...”**
>
>     We appreciate the reviewer’s comment regarding scalability. Our work focuses on domain-specific applications where the number of classes is typically much smaller than in ImageNet. For example, the dermoscopic image dataset contains seven skin conditions (melanoma, melanocytic nevus, basal cell carcinoma, actinic keratosis, benign keratosis, dermatofibroma, and vascular lesions), and the retinal OCT dataset contains four classes (Normal, CNV, DME, and Drusen).
>
>     In these applications, being able to quantify uncertainty, explain predictions, and reliably distinguish between classes is of high practical value. We acknowledge that the class-specific autoencoder architecture may face challenges as the number of classes grows substantially, and scaling to datasets with hundreds or thousands of classes would require additional considerations. Nonetheless, for the targeted domains of this study, the architecture provides meaningful and interpretable uncertainty estimates alongside accurate classification.
>
> **References:**
> 1. Malinin, A. and Gales, M., 2018. Predictive uncertainty estimation via prior networks. Advances in neural information processing systems, 31.
> 2. Durasov, N., Bagautdinov, T., Baque, P. and Fua, P., 2021. Masksembles for uncertainty estimation. In Proceedings of the IEEE/CVF Conference on Computer Vision and Pattern Recognition (pp. 13539-13548).
> 3. Gawlikowski, J., Tassi, C.R.N., Ali, M., Lee, J., Humt, M., Feng, J., Kruspe, A., Triebel, R., Jung, P., Roscher, R. and Shahzad, M., 2023. A survey of uncertainty in deep neural networks. Artificial Intelligence Review, 56(Suppl 1), pp.1513-1589.

---

### Meta-Review · Area_Chair_ij4S · 2025-12-29

**Summary:**

This paper proposes a reconstruction-based uncertainty quantification for multi-class classification.

Apart from reviewer mVt2, most reviewers express substantial concerns regarding scalability, as the method requires training an autoencoder per class. The rebuttal acknowledges this limitation and restricts the intended scope to a few-class settings, which does not convincingly mitigate concerns. The review by reviewer u4aB is brief and thus receives lower weight for this meta-review. Reviewer mVt2 is focused on the relation of the submission to calibration, which the rebuttal partly addresses. They also raise concerns regarding missing experiments for uncertainty explanation, which the authors address in their rebuttal, but only for tabular data and weak baselines.
The strongest objections come from reviewer QBBi, who is highly confident in their assessment. In addition to scalability, they highlight limited novelty, noting similarities with earlier reconstruction-based approaches. The rebuttal attempts to distinguish the method, but the novelty concerns likely remain.

Overall, while the rebuttal satisfactorily addresses some secondary concerns (e.g., missing explanation experiments, clarification of uncertainty decomposition), the central criticism on scalability and limited novelty remains unresolved. Several reviewers acknowledge the potential value of the framework, but believe further development and stronger evaluation are required.

**Reviewer Concerns:**

QBBi Limited novelty: Not convincingly addressed.

QBBi Poor writing: Not convincingly addressed.

QBBi Confusing uncertainty decomposition: Addressed.

QBBi Scalability: Not convincingly addressed.

u4aB Missing comparison to autoencoder-based AD: Partly addressed.

u4aB Scalability: Not convincingly addressed.

mVt2 Missing experiments on epistemic uncertainty: Addressed.

mVt2 Missing comparison to uncertainty calibration methods: Partly addressed.

mVt2 Missing metrics: Addressed.

mVt2 Missing experiments on explanation evaluation baselines: Partly addressed (on tabular data and for weak baselines only).

vGHo Outdated discussion on Bayesian Methods in Section 1.2.1: Addressed.

vGHo The distinction between thetas is not used within the method: Addressed.

vGHo Comparison to OOD: Partly addressed.

vGHo Scalability: Not convincingly addressed.

**Reviewer Scores:**

QBBi: Probably would have stayed with 2.

u4aB: Probably would have raised from 4 to 6.

mVt2: Probably would have raised from 4 to 6.

vGHo: Would have stayed with 4.

---

### Decision · Program_Chairs · 2026-01-26

Reject